# Recent Advances of Immune Checkpoint Inhibition and Potential for (Combined) TIGIT Blockade as a New Strategy for Malignant Pleural Mesothelioma

**DOI:** 10.3390/biomedicines10030673

**Published:** 2022-03-14

**Authors:** Sophie Rovers, Annelies Janssens, Jo Raskin, Patrick Pauwels, Jan P. van Meerbeeck, Evelien Smits, Elly Marcq

**Affiliations:** 1Center for Oncological Research (CORE), Integrated Personalized and Precision Oncology Network (IPPON), University of Antwerp, 2610 Wilrijk, Belgium; annelies.janssens@uza.be (A.J.); patrick.pauwels@uza.be (P.P.); jan.vanmeerbeeck@uza.be (J.P.v.M.); evelien.smits@uza.be (E.S.); elly.marcq@uantwerpen.be (E.M.); 2Department of Thoracic Oncology, Antwerp University Hospital (UZA), 2650 Edegem, Belgium; jo.raskin@uza.be; 3Department of Pathology, Antwerp University Hospital (UZA), 2650 Edegem, Belgium

**Keywords:** mesothelioma, cancer immunotherapy, immune checkpoint blockade, TIGIT, PD-L1

## Abstract

Malignant pleural mesothelioma (MPM) is a fatal cancer type that affects the membranes lining the lungs, and is causally associated with asbestos exposure. Until recently, the first-line treatment consisted of a combination of chemotherapeutics that only had a limited impact on survival, and had not been improved in decades. With the recent approval of combined immune checkpoint inhibition for MPM, promising new immunotherapeutic strategies are now emerging for this disease. In this review, we describe the current preclinical and clinical evidence of various immune checkpoint inhibitors in MPM. We will consider the advantages of combined immune checkpoint blockade in comparison with single agent checkpoint inhibitor drugs. Furthermore, recent evidence suggests a role for T cell immunoglobulin and ITIM domain (TIGIT), an inhibitory immunoreceptor, as a novel target for immunotherapy. As this novel immune checkpoint remains largely unexplored in mesothelioma, we will discuss the potential of TIGIT blockade as an alternative therapeutic approach for MPM. This review will emphasize the necessity for new and improved treatments for MPM, while highlighting the recent advances and future perspectives of combined immune checkpoint blockade, particularly aimed at PD-L1 and TIGIT.

## 1. Introduction

Malignant pleural mesothelioma (MPM) is a fatal cancer that affects the membranes lining the lungs. It is the most common type of mesothelioma (90% of cases), followed by peritoneal mesothelioma [1,2]. It is usually asbestos-related and characterized by a long latency period of 20–40 years, from the first exposure to asbestos fibers, till the development of the disease [1,2,3,4]. Although considered to be a rare disease, MPM incidence has increased in recent years, and is expected to further increase in the following decades due to the widespread use of asbestos up until the late 1990s, and its long latency period [1,4,5]. The prognosis of MPM remains very poor, with a 5-year survival rate of less than 5%, and a median survival of 12–14 months after diagnosis [3,6,7]. Prognostic factors include age, sex, tumor stage, and histology, with significantly worse prognosis associated with the sarcomatoid subtype versus the epithelioid subtype [2,8,9]. Early diagnosis and prompt tumour resection offer the greatest chance of long-term benefits. Unfortunately, MPM is usually identified at an advanced stage, due to its asymptomatic early development and the non-specific nature of its presenting symptoms. In fact, common symptoms presenting at diagnosis include shortness of breath, chest pain, fatigue, and weight loss, at which point, the cancer has usually progressed to become inoperable [10,11]. While a study of diagnostic and prognostic biomarkers of MPM is beyond the scope of this review, it is clear that this merits further exploration for the improvement of the clinical management of the disease.

There is currently no formally acknowledged cure for mesothelioma; treatment is usually aimed at improving patients’ quality of life and extending survival. For many years, the first-line palliative therapy for unresectable MPM consisted of a chemotherapeutic combination of a platinum-salt (cis- or carboplatin) and an antifolate (pemetrexed or raltitrexed). This systemic treatment is generally well-tolerated when folic acid and vitamin B12 supplementation are added to suppress hematological toxicity [3,12,13]. However, the survival benefit is limited, with the EMPHACIS study reporting an increase in median overall survival of only 2.8 months with cisplatin combined with pemetrexed (12.1 months), compared to cisplatin monotherapy (9.3 months) [13]. Moreover, this treatment regimen rarely represents a long-term solution, as about 80% of patients develop recurrent disease within 6–8 months [3]. Furthermore, the addition of the anti-angiogenic agent bevacizumab to cisplatin/pemetrexed therapy was evaluated in the 2016 MAPS trial, where it was found to improve overall survival by only 2.7 months compared to chemotherapy alone (18.8 vs. 16.1 months, respectively) [14]. Thus, although promising, triple cisplatin/pemetrexed/bevacizumab therapy has yet to be globally accepted as the standard of care for mesothelioma. As such, there is an urgent need for novel therapeutic strategies for MPM.

Clinical and preclinical studies have demonstrated that the combined blockade of both the PD-1/PD-L1 and the CTLA-4 immune checkpoints improves anti-tumor immunity compared to the single-agent blockade of either immune checkpoint [15]. However, checkpoint inhibitors aimed at CTLA-4 are more frequently associated with severe immune-related adverse events, which are exacerbated when combined with another checkpoint inhibitor [16]. Therefore, the development of alternative checkpoint blockade strategies to combine with anti-PD-1/PD-L1 warrants further attention. In this review, we study the blockade of the TIGIT immune checkpoint as an interesting potential partner for PD-1/PD-L1 inhibition in MPM.

## 2. Immune Checkpoint Blockade

### 2.1. Background

The field of immunotherapy has certainly revolutionized cancer treatment, so it is no surprise that immunotherapeutic options are also being investigated for MPM. Although MPM is characterized by a low tumor mutational burden and an overall immunosuppressive tumor microenvironment, the latter is also highly inflammatory, due to the presence of asbestos fibers which can invoke a chronic inflammatory response [17,18,19,20]. Indeed, chronic inflammation has long been associated with MPM as a biomarker with prognostic and predictive value, where the type of inflammatory response has a marked impact on MPM patient prognosis [20]. More specifically, it has been shown that the upregulation of the unspecific or innate immune response is coupled with the suppression of the specific or adaptive immune system results in aggressive disease and poor patient outcome [21]. Accordingly, numerous clinical trials investigating various immunotherapeutic strategies have been performed in an attempt to reinvigorate the anti-tumor immune response. Perhaps the most promising immunotherapeutic approach of the past decade has been the use of immune checkpoint inhibitors.

T lymphocytes are capable of recognizing tumor-associated antigens through their T cell receptor (TCR), but ultimately, the T cell’s functional fate depends on its interaction with co-receptors delivering a secondary signal. It is in fact the balance between co-stimulatory and inhibitory signals (immune checkpoints) that determines the strength and duration of the anti-tumor immune response [22]. In normal physiology, these immune checkpoints play a crucial role in the maintenance of self-tolerance, by preventing inappropriate immune reactions and autoimmunity [22,23]. One of the most well-characterized immune checkpoints, programmed death-1 (PD-1), is expressed on activated T cells, macrophages, regulatory T cells (Tregs), and natural killer (NK) cells, while its ligands PD-L1 and PD-L2 are expressed on T cells, dendritic cells (DCs), and a variety of other cell types, including tumor cells [22,24]. Immune checkpoints are often exploited by tumor cells as a mechanism of escape from immune surveillance, as illustrated by the overstimulation of the PD-1/PD-L1 signaling pathway, leading to reduced T cell activation and tumor-specific T cell responses in various cancers [24,25]. Additionally, PD-L1-expressing tumor cells have been found to be intrinsically resistant to T cell cytotoxicity and pro-apoptotic stimuli [24,26]. Immune checkpoint molecules have thus become key clinical targets, leading to the development of checkpoint inhibitors that block these pathways and result in the reactivation of tumor-specific T cells and the restoration of the immune surveillance system [23,24].

In 2018, the Nobel Prize in Physiology or Medicine was awarded to James P. Allison and Tasuku Honjo for the discovery of immune checkpoints, which paved the way for the development of the first-approved checkpoint inhibitors [27,28]. In fact, in 2011, the FDA approved the first immune checkpoint inhibitor for the treatment of melanoma, consisting of a monoclonal antibody (ipilimumab) targeting the well-characterized CTLA-4 immune checkpoint. This was closely followed by the approval of anti-PD-1 antibodies pembrolizumab and nivolumab in 2014 [29]. Various additional antibodies targeting the CTLA-4 and PD-1/PD-L1 immune checkpoints have been developed since, and approved as single agents or combined with other drugs for the treatment of about 50 cancer types [28]. As of 2019, there were over 3000 active clinical trials investigating T cell-targeted immune modulators, making it the largest field in immune-oncological research, after having seen an impressive increase in new targets being studied [28,30].

### 2.2. Immune Checkpoints in MPM

In mesothelioma, chronic inflammation due to asbestos exposure leads to the increased production of free radicals and reactive oxygen species, resulting in the generation of a tumor immune microenvironment (TIME) and the reduction of anti-tumor immunity [31]. While the role of the various cellular and molecular components of the TIME in MPM onset and progression remains to be further elucidated, it does provide a rationale for studying immune checkpoint inhibition as a potential therapeutic option in this disease. In fact, for the first time since the approval of the combination of pemetrexed and cisplatin in 2004, a new first-line treatment was approved for patients with unresectable MPM in 2020: the combination of immune checkpoint inhibitors ipilimumab and nivolumab (CheckMate 743; NCT02899299) [32,33]. This is a very exciting development, as this particular combination has shown very promising results and gained approval in various cancer types, including renal cell carcinoma, microsatellite instability (MSI)-positive colorectal cancer (CRC), and non-small cell lung cancer (NSCLC) (with chemotherapy) [34,35].

This recent approval of combined immune checkpoint blockade (ICB) as a first-line treatment of MPM came after a decade of clinical trials investigating a variety of checkpoint inhibitors, either as monotherapy or as adjuvant to chemotherapy in the first- or second-line setting, as well as combinations of agents targeting different immune checkpoints [36]. Here, we will summarize the clinical trials investigating different immune checkpoint inhibitors in MPM, and draw a comparison between single agent and combined ICB.

In an attempt to move past the limitations of the first-line treatment with cisplatin + pemetrexed, several trials have investigated the combination of this systemic treatment with ICB in the first-line setting. In a single-arm, phase 2 trial (DREAM), Nowak et al. studied the efficacy of adding durvalumab, an anti-PD-L1 antibody, to standard cisplatin + pemetrexed chemotherapy in 54 patients with untreated, unresectable MPM [37]. After 6 months, progression-free survival (PFS) was 57%, with an overall response rate (ORR) of 48%. Immune-related adverse events (irAEs) of grade 3 or higher were recorded in eight patients (15%), and were manageable through high-dose steroids or other immunosuppressive drugs [37]. This study demonstrated the promising potential and acceptable safety profile of the combination of durvalumab with cisplatin + pemetrexed, and has since progressed to a phase 3 trial versus chemotherapy only (DREAM3R). Other promising phase 2 trials investigating the combination of cisplatin + pemetrexed with pembrolizumab (IND-227), nivolumab (NICITA [38]; JME-001 [39]), or durvalumab (PrE0505 [40]) are also currently on the way, and are summarized in Table 1. While a comprehensive analysis of clinical trials investigating the combination of ICB with other agents is beyond the scope of this review, it is worth mentioning that, following the MAPS trial investigating anti-angiogenesis in combination with first-line cisplatin/pemetrexed treatment [14], bevacizumab is now also being investigated in MPM, in combination with PD-L1 checkpoint inhibitor atezolizumab, in the ongoing phase 3 BEAT-meso trial (NCT03762018). In addition, the use of PD-L1 checkpoint inhibition with or without chemotherapy as a neo-adjuvant therapy to MPM surgery is very promising, with several ongoing phase 2 trials and positive preliminary reports on feasibility and safety [41]. Mature data on pathological response and outcome are eagerly awaited.

In addition to the first-line setting, immune checkpoint inhibitors are also being studied in the context of relapsed MPM. Treatment options are currently very limited for relapsed MPM, highlighting the importance of identifying new and efficient therapeutic approaches [42]. The anti-PD-1 antibody pembrolizumab was reported to cause a partial response (PR) of median 12-month duration in 20% of previously treated MPM patients in the phase 1b KEYNOTE-028 trial [43]. Overall, 52% of patients had the stable form of the disease. Moreover, 20% of patients reported grade 3 irAEs, but no treatment-related deaths occurred. In another phase 2 study conducted by researchers at the University of Chicago, pembrolizumab monotherapy was found to cause a PR in 19% of patients with pre-treated MPM with an ORR of 20%, and median PFS and overall survival (OS) of 4.5 and 11.5 months, respectively [44]. In addition, a higher ORR was observed with increasing PD-L1 expression. Aside from pembrolizumab, the PD-1 inhibitor nivolumab has also been tested in the salvage setting in two phase 2 trials, NivoMes [45] and MERIT [46] (Table 1), the latter of which resulted in the approval of nivolumab for the treatment of unresectable recurrent MPM in Japan. In the phase 1b JAVELIN trial, the anti-PD-L1 antibody avelumab was evaluated for its safety and efficacy in patients with previously treated MPM [47]. Here, median PFS and OS were 4.1 and 10.7 months, respectively, with an ORR of 19% in patients with PD-L1-positive tumors (≥5% PD-L1 cut-off).

Based on the results of these completed trials, new phase 3 trials were initiated. In the recent CONFIRM trial, Fennell et al. investigated nivolumab monotherapy in patients with relapsed MPM. Median PFS and OS were 3 and 10.2 months in the nivolumab group vs. 1.8 and 6.9 months in the placebo group, respectively [48]. Furthermore, the PROMISE-meso trial studied the efficacy of pembrolizumab monotherapy versus single-agent chemotherapy in patients with pre-treated MPM. While the ORR was significantly improved in the pembrolizumab arm (22%) compared to the chemotherapy arm (6%), there was no significant improvement in OS for pembrolizumab over chemotherapy [49].

Finally, inhibitors of the CTLA-4 immune checkpoint have also been investigated as monotherapy in relapsed MPM; CTLA-4 inhibitor tremelimumab was studied in a randomized phase 2b study (DETERMINE). However, median OS did not differ significantly between the tremelimumab-treated group (7.7 months) and the placebo group (7.3 months) [50].

While some ICB monotherapies failed to significantly prolong overall survival in patients with pre-treated MPM compared to the control arm [49,50], combinations of multiple ICB agents are still under investigation. In fact, combinations of different checkpoint inhibitors have proven to be more effective than single agent therapies in MPM, as evidenced by the aforementioned approval of ipilimumab plus nivolumab (CheckMate 743) [32], as well as promising early results from a phase 2 study investigating the combination of tremelimumab with durvalumab (NIBIT-MESO-1) [51]. In this trial, 11 out of 40 patients (28%) had an immune-related OR with a median duration of 16.1 months. Median PFS and OS were 5.7 and 16.6 months, respectively [51]. Coupled with an acceptable safety profile, this combination of ICB appeared promising, and merits further investigation. Furthermore, where the CheckMate 743 trial proved the survival benefit of combined ipilimumab and nivolumab compared to cisplatin + pemetrexed chemotherapy (median OS 18.1 months versus 14.1 months) [32], the phase 2 IFCT MAPS2 trial demonstrated and increased ORR for the combination (28%) compared to nivolumab monotherapy (19%) [52].

Checkmate 9LA, combining dual chemotherapy and dual immunotherapy in patients with advanced (irresectable) NSCLC, reported superiority to dual chemotherapy alone with manageable toxicity, and may be an interesting strategy in the combined setting [53].

There is no doubt that the validation of combined immune checkpoint blockade as a first-line treatment for MPM is an important step forward in MPM-related research. However, there are also disadvantages associated with the combination of ipilimumab and nivolumab, notably an increase in irAEs. Clinical trials investigating this treatment regimen in melanoma patients revealed that over half of the patients enrolled in the studies developed treatment-related grade 3 or 4 adverse events [54,55]. In addition, grade 3 adverse events causally associated with ipilimumab plus nivolumab occurred in 34% of MPM patients in the INITIATE trial, where over 90% of patients experienced adverse events of any grade due to treatment [56]. Furthermore, in the aforementioned IFCT MAPS2 trial, 26% of patients in the ipilimumab + nivolumab combination group experienced grade 3–4 toxicities compared to only 14% in the nivolumab monotherapy group [52]. While most irAEs are manageable and can be effectively treated with glucocorticoids, such immunosuppressive drugs also come with their own additional risks [57]. In addition, the efficacy of ICB is highly dependent on tumor immunogenicity, heterogeneity, and complexity, resulting in variable patient responses [58]. As such, currently available immune checkpoint blockade therapies remain limited in their efficacy, in that they benefit only subsets of patients, and can sometimes cause severe immune-related adverse events [28]. Therefore, we are looking towards discovering novel immune checkpoints that could be safely, accurately, and efficiently targeted for cancer treatment. Interestingly, applying the principle of activating immunostimulatory immune checkpoints or blocking immunosuppressive immune checkpoints can also engage innate immunity in addition to adaptive immunity. In this regard, we have shown that MPM fluid samples as part of the TME also contain PD-1-expressing NK cells and PD-L1-expressing DCs and macrophages, although the percentage of positive cells was lower than for PD-1-expressing T cells and PD-L1-expressing tumor cells [59]. Notably, more NK cells than T cells were positive for the immune checkpoint TIM-3 [59]. It is expected that the best immunotherapy results will be obtained if both the innate and adaptive arm of the immune system are engaged by the therapeutic strategy, since the innate immune system recruits and invigorates adaptive immunity, in addition to a positive feedback loop by a bidirectional crosstalk between both types of immunity. In this regard, in addition to the approved ICB for MPM, the T cell immunoglobulin and ITIM domain (TIGIT) is a very interesting candidate.

**Table 1 biomedicines-10-00673-t001:** Clinical trial results of immune checkpoint inhibitors in malignant pleural mesothelioma.

Trial	Phase	Treatment	Primary Endpoint	N° Patients	ORR	mPFS (mo)	mOS (mo)	Status	Ref	Registration Number
**First-Line Setting (with Chemotherapy)**										
CheckMate 743	3	Platinum/pemetrexed +/− nivolumab + ipilimumab	OS	605	40%	6.8	18.1	Completed	[32]	NCT02899299
DREAM3R	3	Platinum/pemetrexed +/− durvalumab	OS	-	-	-	-	Recruiting	[37]	NCT04334759
IND-227	2/3	Cisplatin/pemetrexed +/− pembrolizumab	PFS, OS	126	-	-	-	Active, not recruiting	-	NCT02784171
NICITA	2	Nivolumab, platinum/ pemetrexed	TNT, safety	92	-	-	-	Recruiting	[38]	NCT04177953
JME-001	2	Nivolumab, cisplatin/ pemetrexed	OR	18	77.8%	8.02	20.8	Completed	[39]	UMIN000030892
PrE0505	2	Durvalumab, cisplatin/ pemetrexed	OS	55	56.4%	6.7	20.4	Completed	[40]	NCT02899195
**Second-line (single agent)**										
KEYNOTE-028	1b	Pembrolizumab	OR	25	20%	5.4	18	Completed	[43]	NCT02054806
University of Chicago	2	Pembrolizumab	Predict response	65	7% (<1% PD-L1) 26% (1–49% PD-L1) 31% (>50% PD-L1)	4.5	11.5	Active, not recruiting	[44]	NCT02399371
NivoMes	2	Nivolumab	DCR	34	24%	2.6	11.8	Completed	[45]	NCT02497508
MERIT	2	Nivolumab	OR	34	29%	6.1	17.3	Completed	[46]	JapicCTI-163247
JAVELIN	1b	Avelumab	OR	53	9%	4.1	10.7	Completed	[47]	NCT01772004
CONFIRM	3	Nivolumab	OS	221	11%	3	10.2	Completed	[48]	NCT03063450
		Placebo		111	1%	1.8	6.9			
PROMISE-meso	3	Pembrolizumab	PFS	73	22%	2.5	10.7	Active, not recruiting	[49]	NCT02991482
		Gemcitabine/ vinorelbine		71	6%	3.4	12.4			
DETERMINE	2b	Tremelimumab	OS	382	4.5%	2.8	7.7	Completed	[50]	NCT01843374
		Placebo		189	1.1%	2.7	7.3			
**Second line (combination)**										
NIBIT-MESO-1	2	Tremelimumab + durvalumab	irOR	40	28%	5.7	16.6	Completed	[51]	NCT02588131
MAPS2	2	Nivolumab + ipilimumab	DCR	62	27.8%	5.6	15.9	Completed	[52]	NCT02716272
		Nivolumab		63	18.5%	4	11.9			
INITIATE	2	Nivolumab + ipilimumab	DCR	34	38%	6.2	-	Completed	[56]	NCT03048474

Abbreviations: ORR, Objective response rate; mPFS, median progression-free survival; mOS, median overall survival; TNT, time to next treatment; DCR, disease control rate; irOR, immune-related objective response.

## 3. TIGIT

### 3.1. Background

In recent years, the T cell immunoglobulin and ITIM domain (TIGIT) immune receptor has garnered interest for its role in tumor immunosurveillance. TIGIT is a member of the PVR-like protein family and is expressed on T cells (CD4^+^, CD8^+^, and Tregs) and NK cells. Its structure consists of an extracellular immunoglobulin variable domain, a type I transmembrane domain, and an intracellular domain, with both an immunoreceptor tyrosine-based inhibitory motif (ITIM) and an immunoglobulin tyrosine tail (ITT)-like motif [60,61].

TIGIT has three ligands; (i) CD113, of which the expression is limited to non-hematopoietic tissues, (ii) CD155 and (iii) CD112, mainly expressed on DCs and T cells, and often overexpressed on tumor cells [60]. TIGIT competes with other members of the PVR-like family, CD96 and DNAM-1, for the binding of CD155 on tumor cells, where TIGIT binds to it with the highest affinity (Figure 1). TIGIT can thus inhibit T cell and NK cell functions by delivering a negative signal via CD155 binding, or through interference with DNAM-1/CD155 binding, which otherwise delivers an activating signal, and enhances cytotoxicity. Indeed, TIGIT can either outcompete DNAM-1 for CD155 binding, or it can directly inhibit DNAM-1 by preventing its homodimerization and thus negatively impact its ability to bind CD155 [62]. The interaction of DNAM-1 with CD155 is thought to induce interferon-gamma (IFNγ) production through which NK cells are stimulated, while TIGIT binding to CD155 likely suppresses IFNγ, resulting in the downregulation of NK cells [62]. Moreover, a study conducted by Yu et al. demonstrated that TIGIT/CD155 binding can modulate DC cytokine production, stimulating the secretion of the anti-inflammatory cytokine IL-10, while downregulating the secretion of pro-inflammatory IL-12, thus creating an immunosuppressive environment [61]. Through its expression on immunosuppressive M2 macrophages, TIGIT further influences the innate immune system by inhibiting macrophage-mediated cytotoxicity and pro-inflammatory cytokine release [63].

Furthermore, CD112 and CD113 are also capable of delivering inhibitory signals to T and NK cells upon TIGIT binding, although these interactions are not as strong as TIGIT/CD155. In addition, both TIGIT and DNAM-1 compete for binding CD112 with the CD112 receptor (CD112R), which also acts as an inhibitory immune checkpoint [60].

TIGIT is only weakly expressed in naïve cells, but is known to be upregulated in T and NK cells upon activation. Consequently, TIGIT was found to be highly expressed on tumor-infiltrating lymphocytes (TILs), and high levels of TIGIT expression have been associated with tumor progression [60,62,64,65]. In addition, several studies have investigated the role of TIGIT expression on Tregs. Kurtulus et al. showed that the majority of CD4^+^ TILs expressing TIGIT were in fact FOXP3^+^ Tregs [66], which were previously shown to specifically inhibit pro-inflammatory Th1 and Th17 cell responses, but not Th2 [62,67]. Indeed, in vivo experiments revealed that TIGIT-positive Tregs in the TIME displayed a much more suppressive phenotype compared to TIGIT-negative Tregs [66]. The fact that TIGIT expression is also associated with severely dysfunctional cytotoxic tumor-infiltrating T cells, and that their functionality was essentially restored in the absence of TIGIT in preclinical cancer models, serves as further evidence of TIGIT-mediated immunosuppression [66].

### 3.2. Preclinical and Clinical Results with Single-Agent TIGIT Blockade

The TIGIT/CD155 signaling pathway thus plays an important role in anti-tumor immunity, as it acts as an immune checkpoint capable of suppressing tumor-specific cytotoxic T/NK cell responses. It thus comes as no surprise that various monoclonal antibodies targeting the TIGIT immune checkpoint are currently in advanced stages of clinical development for solid tumors (Table 2) [70]. Similar to ipilimumab and nivolumab, most anti-TIGIT antibodies currently under investigation are fully human, and therefore have a very low risk of unwanted immunogenicity. The majority of TIGIT-inhibitor antibodies also have an IgG1 backbone, resulting in a significant antibody-dependent cellular cytotoxicity (ADCC) response from innate immune cells. While some anti-TIGIT antibodies have an active FCγ-receptor (FCγR) binding region which is inactive in others, the effect of FCγR status on the clinical efficacy of the antibodies remains to be elucidated [70].

Several preclinical and clinical studies have shown that the blockade of TIGIT using antagonistic monoclonal antibodies can reverse cytotoxic T lymphocyte exhaustion and enhance NK cell killing. Firstly, Guillerey et al. evaluated the therapeutic potential of TIGIT blockade to reinvigorate the anti-tumor immune response against multiple myeloma (MM) in the preclinical and clinical setting [64]. In mice bearing MM tumors, TIGIT expression in CD8^+^ T cells correlated with myeloma burden, and was detected on 30–40% of FOXP3^+^ Tregs. TIGIT was also found to be more highly expressed on CD8^+^ T cells in the bone marrow of newly diagnosed or relapsed MM patients compared to that of healthy donors, and was associated with higher PD-1 expression. In addition, these TIGIT-positive CD8^+^ T cells had decreased cytokine production, cytotoxic, and proliferative capacities, indicating that this subset of cells is largely functionally exhausted. Following TIGIT blockade in MM-bearing mice, tumor burden was significantly reduced, and survival was prolonged in a CD8^+^ T cell-dependent manner. CD8^+^ T cell effector functions appeared to be restored as cytotoxicity and cytokine production significantly improved.

In a 2018 study by Zhang et al., the link between TIGIT expression and NK cell exhaustion was examined [65]. TIGIT expression was shown to be significantly higher on NK cells derived from the intratumoral region of patients with colon cancer, compared to NK cells from the tumor-surrounding tissue. In contrast, TIGIT expression on CD8^+^ T cells did not differ significantly between cells from either region. TIGIT expression was also observed on tumour infiltrating CD8^+^ T cells and NK cells in mouse models of melanoma, colon cancer, breast cancer, and fibrosarcoma [65]. TIGIT-expressing NK cells possessed reduced effector functions and tumor-killing capabilities, as became evident by their reduced expression of cytokines such as IFNγ and tumour necrosis factor (TNF). The CD155 ligand was also abundantly expressed within the TIME of both human and murine tumors, resulting in the exhaustion of TIGIT-positive NK cells. They also observed that genetic TIGIT deficiency, as well as NK cell specific TIGIT deficiency, both led to improved overall survival, as well as repaired CD8^+^ and NK cell effector functions in a mouse melanoma model. TIGIT blockade using a monoclonal antibody resulted in delayed tumor growth, reduced metastasis, improved survival, and the reversed exhaustion of tumor-infiltrating NK cells in preclinical models of colon cancer, breast cancer, and fibrosarcoma [65].

In 2020, Maas et al. investigated the role of the TIGIT pathway in the NK-mediated anti-tumor immune response against ovarian cancer [71]. Their results indicated a significant reduction in DNAM-1 expression on peritoneal NK cells from ovarian cancer patients compared to healthy donor NK cells (51.8% vs. 90.9%, respectively), whereas there was no significant difference in TIGIT expression. On the other hand, CD96 expression was significantly higher in patient-derived NK cells in comparison to healthy donor NK cells (84.2% vs. 44.1%, respectively). These data point towards a shift in co-receptor expression from the activating DNAM-1 receptor towards the inhibitory receptor CD96, thus creating a more exhausted NK cell subpopulation. Treatment with a TIGIT inhibitor caused reduced tumor growth and improved NK cell functionality, both in mouse models of ovarian cancer and patient-derived cell cultures.

Park et al. previously presented the results of their studies on anti-TIGIT antibodies capable of blocking ligand binding and TIGIT signaling [72]. Their antibody targeting mouse TIGIT (313R12) was shown to have positive effects on murine models of colon and kidney cancer through tumor growth inhibition. Interestingly, the TIGIT blockade appeared to have a lasting effect on immune memory, as re-challenge failed to generate new tumors in some animals. In addition, they demonstrated that the depletion of both CD4^+^ and CD8^+^ T cells severely impacted the efficacy of TIGIT blockade, indicating that these cell populations are critical for the immune-related effects of anti-TIGIT.

### 3.3. Combining TIGIT Blockade with Other ICB

Despite the promising results obtained with TIGIT blockade in various cancers, anti-TIGIT monotherapy has been reported to be insufficient to cause the regression of already established tumors in mouse models of colon carcinoma and glioblastoma [73,74]. In addition, no objective responses were observed following single-agent TIGIT blockade in patients with multiple advanced solid tumor types, including NSCLC [75,76].

In a preclinical study, Johnston et al. found that single-agent TIGIT blockade failed to delay tumor growth or significantly prolong survival in mice bearing CRC tumors [73]. In this study, TIGIT involvement in anti-tumor CD8^+^ responses was investigated, and TIGIT was shown to be highly expressed on TILs in human lung squamous cell carcinoma, colon adenocarcinoma, uterine corpus endometroid carcinoma, breast cancer, and renal clear cell carcinoma. Interestingly, TIGIT expression was found to correlate with CD8 and PD-1 expression in both human and mouse CRC. Both anti-TIGIT and anti-PD-L1 agents were subsequently tested in a mouse model of CRC, where neither monotherapy managed to have a significant effect on tumor growth and survival. However, the combined blockade of TIGIT and PD-L1 resulted in an impressive 75% decrease in tumor volume, with the majority of mice achieving a complete response and a 75% survival rate (compared to 10% for anti-TIGIT monotherapy). In addition, tumor antigen-specific immune memory appeared to have been induced by the combination therapy, as rechallenge with CT26 CRC cells did not result in new tumor growth.

Similar results were obtained by Dixon et al., who reported that, while anti-TIGIT monotherapy did cause a small delay in murine colon carcinoma tumor growth, its combination with anti-PD-1 therapy resulted in complete tumor regression in all mice [74]. Tumor-infiltrating CD4^+^ and CD8^+^ T cell functionality was markedly improved only in the combination group, not the monotherapy groups, as shown by increased cytokine production. PD-L1 and TIGIT co-blockade also led to a significant survival benefit in a glioblastoma model, with 17% of mice showing long-term survival [74].

A simultaneous blockade of TIGIT and PD-1 was also investigated in a study by Chauvin et al., where it was shown to cause a 2.3-fold increase in tumor antigen-specific CD8^+^ T cells compared to IgG control, or either monotherapy [77]. Furthermore, in the 2018 study by Hung et al., the therapeutic effect of combined PD-1 and TIGIT blockade was examined in a murine glioblastoma model [78]. After demonstrating the increased expression of both PD-1 and TIGIT on TILs compared to spleen lymphocyte populations, the effect of combined checkpoint blockade on survival was evaluated. They reported a median survival of 28 days for animals treated with anti-TIGIT monotherapy, which did not differ significantly from the control group. However, all groups receiving combination therapy had significantly prolonged survival, with one group even attaining 85.7% long-term survivors (>80 days). Again, immune memory was established in these long-term survivors, as demonstrated by the 100% survival rate 90 days after re-challenge. Interestingly, this study also found that the co-blockade of TIGIT and PD-1 reduced the number of tumor-infiltrating DCs in this model. In fact, the presence of tumor-infiltrating DCs may contribute to an immunosuppressive tumor microenvironment, and indeed, untreated mice were found to have a significantly larger number of infiltrating DCs compared to mice treated with the combination, while neither inhibitor alone managed to reduce DC infiltration.

In their 2020 paper, Ma et al. explain their slightly different approach to a dual TIGIT and PD-L1 blockade [79]. Rather than using two separate monoclonal antibodies to block both checkpoints, they used nanobody screening technology to develop a multivalent bispecific antibody targeting both TIGIT and PD-L1. This technique is thought to have some advantages over “classic” monoclonal antibody targeting, including a higher binding specificity and affinity, reduced costs, and enhanced tumor killing [80]. The bispecific antibody was shown to enhance T cell effector functions (PBMC IL-2 cytokine secretion) compared to each separate nanobody, thus proving it to be a functional bispecific antibody that merits further study in vivo.

As per the aforementioned preclinical studies, the combined blockade of TIGIT and other immune checkpoints, specifically PD-1/PD-L1, appears to be able to overcome the limitations of single agent therapy, and even has the potential to result in complete tumor regression with the induction of a durable anti-tumor immune memory response [60,73,74]. While the TIGIT immune checkpoint is still a relatively new target, some clinical trials investigating its therapeutic potential alongside PD-1/PD-L1 blockade are ongoing.

In 2020, a clinical trial evaluating anti-TIGIT (tiragolumab) and anti-PD-L1 (atezolizumab) in PD-L1-positive NSCLC was the first to publish results. This phase 2 study by Rodriguez-Abreu et al. (CITYSCAPE; NCT03563716) reports a clinically significant improvement in overall response rate for the combined treatment compared to single agent anti-PD-L1 (31.3% vs. 16.2%, respectively) [81]. Median progression-free survival was also significantly improved for the combination group (5.4 months) in comparison with the atezolizumab monotherapy group (3.6 months). The safety profile was also deemed tolerable, with treatment-related adverse events reported in 72% and 80.6%, of which 19.1% and 14.9% were of grade 3 or higher for the monotherapy versus the combination group, respectively. In fact, tiragolumab is the first anti-TIGIT agent to be granted FDA Breakthrough Therapy Designation, in combination with atezolizumab, as a first-line treatment for metastatic NSCLC. As a result, tiragolumab is now being investigated in combination with atezolizumab in a broad range of cancer types, including head and neck squamous cell carcinoma (NCT04665843), esophageal cancer (NCT04543617), and cervical cancer (NCT04300647) in the SKYSCRAPER trials.

### 3.4. Current Status and Future Perspectives of TIGIT Blockade in MPM

Despite the extensive evidence of the potential of TIGIT blockade in a variety of human solid tumors, TIGIT is yet to be comprehensively investigated in MPM. At this time, data on TIGIT in mesothelioma remain very limited in literature. In what appears to currently be the only study pertinent to this subject, Klampatsa et al. performed an analysis of MPM TILs, which generated some interesting data on TIGIT expression [82]. They found that, in comparison with the immune cell population of tumor-free lungs (TFL), MPM TILs had a much higher number of Tregs (2.2% vs. 12.8% of CD4^+^ population, respectively), and that these highly expressed TIGIT (72.5% of cells). In addition, TIGIT was also highly expressed on MPM CD8^+^ TILs (58.7%) compared to CD8^+^ cells from TFLs (33.4%). TILs expressing TIGIT were also found to secrete significantly less IFNγ (8.3%) compared to TIGIT-negative cells (15%). TIGIT thus appears to mark hypofunctional TILs in MPM, caused both by the presence of TIGIT-positive Tregs, as well as TIGIT expression on tumor-infiltrating CD8^+^ T cells.

In addition, a dose-finding study in NSCLC patients found that anti-TIGIT monoclonal antibodies are generally well tolerated, with no grade 3–4 adverse events for monotherapy, and grade 3–4 irAEs occurring in only 10% of patients treated with anti-TIGIT combined with pembrolizumab [70,83]. This, alongside the favorable toxicity profile of anti-PD-1/PD-L1 compared to anti-CTLA-4 [16], suggests that the treatment-related adverse effects of TIGIT and PD-1/PD-L1 co-blockade in solid tumors (including MPM) would be manageable. Furthermore, we have previously confirmed PD-1/PD-L1 immune checkpoint expression [59] and the efficacy of anti-PD-L1 treatment [84] in MPM. Taken together, these findings suggest that PD-1/PD-L1 and TIGIT co-blockade may be a good strategy for the immunotherapeutic treatment of MPM that warrants further investigation. It is evident from the extremely scarce information on TIGIT in the MPM in the literature that additional studies are urgently needed to get a clearer view on the potential of TIGIT blockade in this disease, let alone its combination with the PD-1/PD-L1 blockade.

Indeed, the most effective treatment regimen including the TIGIT blockade remains to be elucidated. Considering the increased risk of severe immune-related adverse events observed with combined ICB, particularly when an anti-CTLA-4 agent is used, it would probably not be optimal to add an anti-TIGIT agent to the newly approved ipilimumab + nivolumab combination. Researchers currently aim to replace anti-CTLA-4 agents, for which anti-TIGIT is a good candidate for further investigation. As for the standard chemotherapeutic combination of cisplatin + pemetrexed, many (combined) immune checkpoint inhibitors have been shown to synergize with this (Table 1). We previously found no counterindication to combining ICB with first-line chemotherapy in MPM [85], suggesting that adding an anti-TIGIT agent to this combination could also have a synergistic effect, particularly in the neo-adjuvant setting, as we showed that cisplatin could potentially downregulate immune checkpoint expression on MPM cells [85]. Also, radioimmunotherapeutic strategies are being investigated in preclinical and clinical studies, and radiation therapy has been shown to be linked to higher overall survival rates in MPM patients [86]. However, the optimal conditions for combining radiation with immune checkpoint inhibitors remain to be determined. Currently, a phase I clinical trial in MPM patients is ongoing to determine the feasibility of combining stereotactic body radiation therapy with immune checkpoint inhibition (NCT04926948), the results of which will surely benefit any future studies investigating TIGIT blockade in combination with radiotherapy.

## 4. Conclusions

It is clear that the exceptionally poor prognosis of MPM warrants further research into more effective treatments for this disease. The approval of nivolumab + ipilimumab as a first-line therapy represents a turning point for immune checkpoint blockade in mesothelioma, with multiple checkpoint inhibitors being investigated in clinical trials, in an attempt to improve upon existing therapies. However, despite its promising results and various ongoing advanced stage clinical trials in multiple solid malignancies, a blockade of the TIGIT immune checkpoint is yet to be extensively evaluated in MPM. While the data summarized in this review suggest that TIGIT blockade and its combination with anti-PD-(L)1 may hold promise as a new and effective immunotherapeutic treatment for MPM, further preclinical and clinical studies are necessary to fully investigate its potential.

## Figures and Tables

**Figure 1 biomedicines-10-00673-f001:**
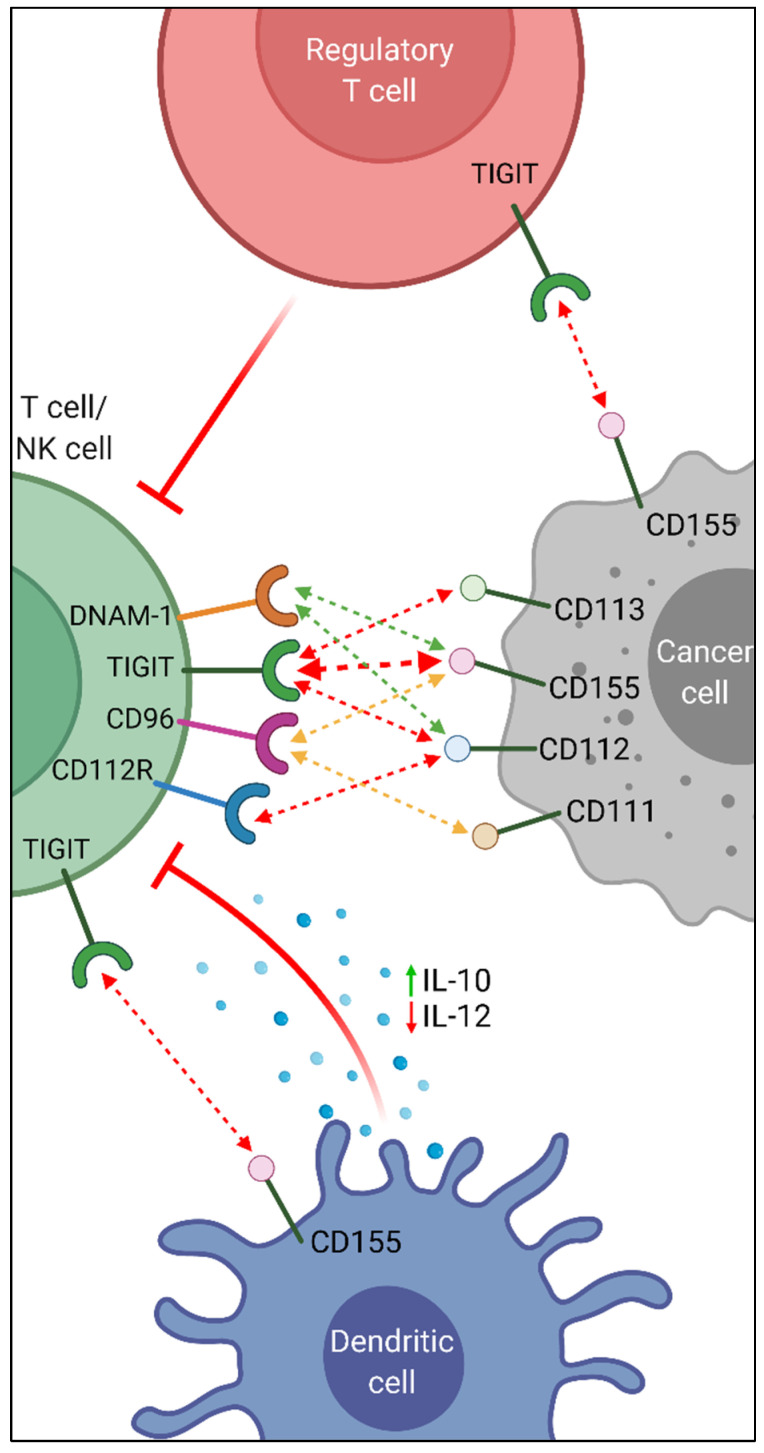
T cell immunoglobulin and ITIM domain (TIGIT) signaling pathway and mechanism of action. TIGIT, DNAM-1, CD96, and CD112R are mainly expressed on NK cells and T cells, including Tregs. Their ligands CD155, CD111, CD112, and CD113 are expressed on cancer cells and antigen-presenting cells such as DCs. Upon ligand binding, TIGIT and CD112R deliver negative signals (red arrows) to cells through their cytoplasmic regions, thereby either directly inactivating immune cell effector functions, or indirectly through Treg-mediated suppression. CD155 also delivers an inhibitory signal to cells when bound, resulting in another cell-extrinsic mechanism of immune suppression through increased anti-inflammatory and decreased pro-inflammatory cytokine secretion. DNAM-1 delivers a positive signal to cells when bound to CD155 or CD112 (green arrows), thus promoting anti-tumor immunity. However, this interaction is often outcompeted by TIGIT and CD112R which bind these ligands with higher affinity. CD96 has an inhibitory role in mice by negatively controlling NK cell cytokine responses [68]. However, the role of CD96 in humans remains ambiguous with both activating and inhibitory functions (orange arrows) having been reported [69]. Created with BioRender.com.

**Table 2 biomedicines-10-00673-t002:** Anti-TIGIT monoclonal antibodies in phase II/III of clinical development.

Antibody	Type	FcγR Status	Company	Clinical Trial Name	Clinical Trial Treatment	Cancer Type	Phase	Status	Registration Number
Tiragolumab	Fully human IgG1	Active	Genentech	CITYSCAPE	Tiragolumab + atezolizumab	NSCLC	2	Active, not recruiting	NCT03563716
				SKYSCRAPER-01	Tiragolumab + atezolizumab	NSCLC	3	Recruiting	NCT04294810
				SKYSCRAPER-02	Tiragolumab +/− atezolizumab, carboplatin, etoposide	SCLC	3	Active, not recruiting	NCT04256421
				SKYSCRAPER-07 (Hoffman-La Roche)	Atezolizumab +/− tiragolumab	Oesophageal SCC	3	Recruiting	NCT04543617
Ociperlimab	Humanized IgG1	Active	BeiGene USA, Inc.	AdvanTIG-202	Tislelizumab (anti-PD-1) +/− ociperlimab	Cervical cancer	2	Active, not recruiting	NCT04693234
				AdvanTIG-203	Tislelizumab +/− ociperlimab	Oesophageal SCC	2	Recruiting	NCT04732494
				AdvanTIG-302	Ociperlimab + tislelizumabvs pembrolizumab	Lung cancer	3	Recruiting	NCT04746924
				-	Ociperlimab + tislelizumab + chemoradiotherapy	SCLC	2	Recruiting	NCT04952597
Vibostolimab	Fully human IgG1	Active	Merck & Co Inc.	KEYMAKER-U01	Pembrolizumab + chemo + vibostolimab	NSCLC	2	Recruiting	NCT04165070
				KEYMAKER-U02	Pembrolizumab + vibostolimab	Melanoma	2	Recruiting	NCT04305054
Domvanalimab	Fully Human IgG1	Inactive	Arcus Biosciences Inc.	ARC-7	Domvanalimab +/− zimberelimab/etrumadenant	NSCLC	2	Recruiting	NCT04262856
				-	Domvanalimab + zimberelimab	Melanoma	2	Not yet recruiting	NCT05130177
BMS-986207	Fully human IgG1	Inactive	Bristol-Myers Squibb Co.	-	BMS-986207 + ipilimumab + nivolumab	NSCLC	2	Not yet recruiting	NCT05005273
EOS-448	Fully human IgG1	Active	iTeos Therapeutics SA	TIG-006	EOS-448 + pembrolizumab vs. EOS-448 + inupadenant	Advanced solid tumours	2	Recruiting	NCT05060432

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
