# Peer review of "Recent Advances of Immune Checkpoint Inhibition and Potential for (Combined) TIGIT Blockade as a New Strategy for Malignant Pleural Mesothelioma"

_biomedicines, 2022, doi:10.3390/biomedicines10030673_

Round 1

Reviewer 1 Report

The current preclinical and clinical evidence of various immune checkpoint inhibitors for the treatment of patientes suffering from malignant pleurale mesothelioma were well summarized in this review. The efficacy and safety of the combined immune checkpoint blockade (e.g. anti-PD-1- or anti-PD-L1- antibodies plus anti-CTLA4-antibodies) were compared with single agent checkpoint inhibition (e.g. anti-PD-1- or anti-PD-L1- antibodies such as avelumab, nivolumab or pembrolizumab or anti-CTLA4-antibodies such as tremelimumab) were compared in detail. Later, the authors explain in detail why inhibition of T cell immunoglobulin and ITIM domain (TIGIT) in addition to PD-1- or PD-L1-inhibition might be more effective than single-agent inhibition of the PD-1- or PD-L1-inhibition why TIGIT-inhibition might act as a novel target for immunotherapy for the treatment of  malignant pleural mesothelioma. Unfortunately, no clinical phase 2 or 3 studies are investigating TIGIT-inhibition in malignant pleural mesothelioma.

Maybe the title of the review might be reworded in: "Recent advances of immune checkpoint inhibition and potential for (combined) TIGIT blockade 2
as a new strategy for malignant pleural mesothelioma" because the long introduction highlights the results of immune checkpoint inhibition clinical studies in malignant pleural mesothelioma.

Reviewer 2 Report

In the present manuscript “Recent advances and potential for (combined) TIGIT blockade as a new strategy for mesothelioma”, Rovers et al summarizes the most recent advances regarding immune therapy in malignant mesothelioma with a special focus on TIGIT treatment.

This is an interesting manuscript and malignant pleural mesothelioma still strongly deserves novel therapy approaches to improve its dismal prognosis. However, some major issues have to be addressed before publication can be recommended from my side.

Line 57: you mention today’s MPM standard therapy, namely cis/pem, however, more recently also bevacizumab has been widely accepted as third agent added to the cis/pem therapy. Can you also comment on this since antiangiogenic therapy might also interfere with immune therapy?

In the review you mainly focus on the effect and role of the specific immune system, however, the innate immune system also proved to play a relevant role as demonstrated and reviewed before. Can you also discuss this point and give a perspective if this part of our immune system might also as well serve as potential therapy target in the future? The duality of our immune system in MPM development and progression has been analyzed and reviewed before and should be further discussed. Please also compare “Inflammation in malignant mesothelioma—friend or foe” by Linton et al and “Biomarkers for Malignant Pleural Mesothelioma—A Novel View on Inflammation” by Vogl et al. Do you think, TIGIT is also influencing TAMs and their phenotype? Do you know, if TIGIT also influences the innate inflammatory response?

Can you also comment on the potential interesting setting of (in general) immune therapy within multimodality therapy approaches including macroscopic radical surgery? What is your expectation: do you think, immune therapy will rise the number of patients that will benefit from macroscopic radical surgery or does this mean that immune therapy in the end might “replace” mesothelioma surgery. This would be an interesting issue at least from the surgical point of view and should be addressed in the discussion.

When talking about the best combination and most effective treatment regimen, do you think, that a triple checkpoint inhibitor therapy might make sense (CTLA4, PDL1 and TIGIT)? What about combining with chemotherapy and also with radiotherapy (which is known to influence the immune response during immune therapy- abscopal effect)? Please also discuss this issue at least for first line/intended for induction chemotherapy patients.

Reviewer 3 Report

The review "Recent advances and potential for (combined) TIGIT blockade as a new strategy for mesothelioma" is well written and I have only one minor comment

  • The fact is data on TIGIT in mesothelioma remains very limited in the literature. Improve the discussion further based on this fact
